# Chronic Kidney Disease of Uncertain Etiology in Sri Lanka: Curing between Medicine and Traditional Culture

## Chandani Liyanage

Department of Sociology, Faculty of Arts, University of Colombo, P.O. Box 1490, Colombo 03, Sri Lanka; chandaniliyanage@soc.cmb.ac.lk

**Abstract:** Chronic Kidney Disease of unknown origin (CKDu) has appeared across Sri Lanka's North Central Province (NCP) since the 1990s as an epidemic, unexplained by conventional associated risk factors. During the past few decades, a large number of studies attempted to determine the unknown etiology of CKDu. Despite these investigations, no concrete conclusions were developed, though a number of contradictory hypotheses emerged. The present ethnographic study was carried out in two endemic areas, labelled as "CKDu hotspots", and illuminates how curing takes place between biomedicine and traditional cultural practices. Our ethnographic study thoroughly scrutinized three decades of lived experience, lay-perceptions and local discourses on CKDu. We used a qualitative study design with a transcendental phenomenological approach and employed a mixture of ethnographic methods. Data collection techniques included participant observation, in-depth interviews, focus group discussions and key informant interviews. Data was analysed by using an interpretive thematic analysis model. Findings revealed that lay people have constructed a popular discourse on CKDu, and we explored their views on the origin, etiology and prevalence of CKDu in their locality over the past few decades. Patients' narratives revealed that there were currently a number of gaps in service delivery. These were mainly due to distant relationships between healthcare providers and CKDu patients. Lay people in affected communities were marginalized throughout the investigation process to determine the unknown etiology, their involvement marginalized to merely acting as objects for scientific instigation. The affected communities strongly believed that CKDu was a recent phenomenon resulting from the mismanagement of the natural environment due to social and lifestyle changes. These findings highlight local dynamics of healthcare seeking behaviours which demand complementary medicine system, particularly given the number of limitations in the biomedical system. Empirical evidence generated from this study suggests a conceptual shift to an ethno-medical model to address CKDu. Improving cultural competency and communication skills among healthcare providers in public health are crucial in order to apply a "bio-psychosocial perspective" in healthcare delivery system and bridging the gap between hospital and the community.

**Keywords:** cultural practices; popular discourse of CKDu; complementary medicine; bio-psychosocial approach

## 1. Introduction

Chronic Kidney Disease of unknown origin (CKDu) has appeared across Sri Lanka's North Central Province (NCP) since the 1990s as an epidemic, unexplained by conventional associated risk factors such as hypertension and diabetes (Abeysekera et al. 1996; Athuraliya et al. 2009; Athuraliya et al. 2011; Jayatilake et al. 2013; Rajapakse et al. 2016; Ranasinghe et al. 2019). The number of those affected and dying has shown a progressive increase, threatening the existence of entire communities living in these areas. Repeatedly, the NCP has recorded the highest number of deaths due to chronic renal failure in the country (Jayatilake et al. 2013; Wimalawansa 2015; Ranasinghe et al. 2019). Some estimates illustrate that the incidence of CKDu in Sri Lanka has been doubling every four to five years. The number of people affected was approximately 150,000 by 2015 and nearly 3% of

affected people die annually (Wimalawansa 2015). The disease is prevalent among farming communities and has devastated the economy of these areas in a number of ways. The loss of productivity from the illness, the cost of care for those affected, and preventive action have taken a toll on the members of these communities, some of whom were merely surviving in a subsistence economy (WHO (World Health Organization) 2016). Since Sri Lanka lacks a comprehensive psycho-social support system, the loss of productivity, the costs of care and preventive measures have cumulative impacts on affected communities while pushing many toward a new type of poverty situation.

The term CKDu was initially used in El Salvador in the early 2000s to describe Chronic Kidney Disease with unknown/uncertain etiology that predominantly affected agricultural communities from large scale plantations (Lunyera et al. 2016). Subsequently, a similar pattern of disease was reported in other countries including Nicaragua, Costa Rica and Sri Lanka (Gunasekara et al. 2020; Wesseling et al. 2015; Chandrajith et al. 2011). A systematic review of CKDu literature suggests that the disease was identified mainly in agricultural communities in those regions and most conspicuously among young male farmers (Lunyera et al. 2016). While increases of CKDu have also been reported from other Asian countries such as Bangladesh and India (John et al. 2021; Abraham et al. 2015), CKDu in Sri Lanka has attracted significant levels of medical and scientific attention in the past few decades (Jayatilake et al. 2013; Dharma-Wardana et al. 2014; Jayasekara et al. 2015; Jayasumana et al. 2015a, 2015b; Chandrajith et al. 2011; Bandara et al. 2008; Ranasinghe et al. 2019). According to the main findings of these studies, alleged risk factors mainly included agricultural toxins—chemical fertilizers, herbicides and insecticides. Due to increasing evidence that agrochemicals may have caused CKDu, some scholars refer to it as Chronic Agrochemical Nephropathy (Jayasinghe 2014). Later, it was renamed to Chronic Intestinal Nephritis in Agricultural Communities (Jayasumana et al. 2016). The natural hardness of the water that is frequently consumed and prolonged dehydration are also considered as causal factors that contribute toward the pathogenesis of CKDu. The latter suggestion came from the results of studies done in Nicaragua (Wesseling et al. 2016).

Most of the studies done on CKDu in Sri Lanka seem to emphasize the clinical course/s of the disease, signs and symptoms, role of extensive use of agrochemicals and other related environmental risk factors such as hardness of water and contributory factors related to anions and cations. Findings from diverse studies on CKDu carried out during the past three decades in Sri Lanka reveal that there are no concrete conclusions with regard to unknown/uncertain etiology though a number of contradictory hypotheses developed as an outcome of those investigations (Rajapakse et al. 2016). A systemic review of existing literature on CKDu in other affected regions noted similar causative factors including environmental factors (heavy metal exposures, high seasonal temperatures), agrochemical use, contaminated water supplies, snake bites and herbal use, and these have all been studied as potential causes of CKDu while exploring demographic characteristics such as age, sex and occupation as common features among affected communities (Lunyera et al. 2016). Despite the increasing global awareness of CKDu, many risk factors remain uncertain, though agrochemical use and heavy metal exposure were studied most frequently (John et al. 2021; Paidi et al. 2021). Irrespective of many biomedical science-related studies, sociological and anthropological enlightenment on CKDu remains limited, yet recent social scientific scholarship indicates important contextual relationships between CKDu and poverty, chemically-intensive agriculture, experience of structural violence, illness burden on affected communities and impacts of globalization (Bandarage 2013; de Silva et al. 2017; Liyanage 2019; Silva 2020). In spite of the diverse studies on susceptible causative factors of CKDu and common features of demographic variables among affected regions, very few qualitative studies explore the lived experience and socio-cultural aspects of CKDu (John et al. 2021; Paidi et al. 2021; Lunyera et al. 2016). The present ethnographic inquiry, carried out in two endemic areas labelled as "CKDu hotspots" in the North Central Province in Sri Lanka, illuminates how curing takes place between medicine and traditional cultural

practices. Our ethnographic study thoroughly scrutinized three decades of lived experience, lay-perception and local discourse on CKDu.

*Medical Pluralism in Sri Lanka*

Medical pluralism in Sri Lanka is well documented and includes the biomedical/Western medicinal system, traditional medical systems and a huge variety of religious, ritual and magical practices (Obeyesekere 1998; Uragoda 1987; Silva et al. 1994; Liyanage and Ekanayaka 2018). The traditional medical system, commonly known as "*Desiya chikitsa*", is indigenous to Sri Lanka, while Ayurveda and Siddha medical systems came to Sri Lanka from India and the Unani system came to Sri Lanka from the Arab world through trade relationships (Uragoda 1987). Among the traditional medical systems, Ayurveda and "*Desiya chikitsa*" are the two main systems most widely practiced in rural Sri Lanka (Liyanage and Ekanayaka 2018; Obeyesekere 1998).

The earliest system of medicine before the advent of Ayurveda was "*Desiya chikitsa*" or "*Sinhala vedakama*" which was passed from generation to generation as family tradition. The knowledge system of "*Desiya chikitsa*" is based on the experiences of the originators of that particular tradition (the prescription and methods of its application) and then passed down to the next generation in the family. Thus, the healing process is not guided by any philosophical or conceptual framework. This system was officially recognized when the Ayurveda Act was introduced in 1961. It was integrated to a large extent with Ayurveda, though some healers practice it in its pure form within their family traditions. However, Ayurveda has a philosophical basis in order to maintain health/well-being while managing ill-health that includes the "five elements of the universe", "three humors" and "seven components' of the body". Yet, in practice, "*Desiya chikitsa*", Siddha and Unani medical systems have all been largely integrated with the Ayurveda medical system (Wanninayake 1982). Homeopathic and Chinese medical systems are also practiced in Sri Lanka to some degree, though these two systems occupy limited space within the medical pluralism in Sri Lanka (Liyanage and Ekanayaka 2018).

Biomedicine is the dominant system within the healthcare delivery system in Sri Lanka, which developed during the British colonial period as a vertical structure, a hegemonic and intrusive system that was insensitive to the dynamics of traditional culture (Jones 2004). Margaret Jones, in her analysis on "Health policy in Britain's Model Colony, Ceylon 1900–1948", argued that the health services of Ceylon should be understood within its political, social and economic context. Thus, at the beginning of the 20th century, the solutions applied to health problems in the colonies appeared to be straight forward transfers from the metropolitan centre to the periphery which included the passing of public health legislation such as vaccinations, notification of diseases, medical registration, sanitary laws and so on which were modelled in form and content on similar British legislation (Jones 2004). The vertical healthcare structures established were urban and hospital based, expensive and insensitive to the needs of the indigenous populations and culturally hegemonic (Jones 2004). In practice, equal status was not given to indigenous practitioners and the traditional medical systems slowly became marginalized within the healthcare delivery system in Sri Lanka (Hettige 1991; Uragoda 1987). Thus, the two systems—modern and traditional medical systems—maintained a distance without constructive interaction in healthcare provision (Liyanage and Ekanayaka 2018; Jones and Liyanage 2018). The rich and diverse traditional knowledge system received hardly any policy and programme support from the state or public bodies (Balasubramanian and Devi 2006).

The current dominant medical system of biomedicine within healthcare delivery fails to accommodate psycho-social and cultural aspects of contemporary health hazards. Singer explores how a syndemic model is significant in that it focuses on the bio-social complex, which consists of interacting co-present or sequential diseases and the social and environmental factors that promote and enhance the negative effects of disease interactions (Singer et al. 2017). Empirical evidence further reveals how structural interventions within the biomedical setting can have a greater impact on health than conventional clinical

interventions in disease control (Farmer et al. 2006). This paper analyses curing between medicine and traditional culture with a focus on issues related to the process of identifying a disease with an unknown etiology, the popular discourses of CKDu that are constructed by affected communities, local dynamics of healthcare seeking behaviour and a transformation of "CKDu hotspots" into sites of humanitarian concerns. Our ethnographic study critically evaluates strengths and gaps in the healthcare delivery system and explores how the bio-psychosocial approach could be incorporated into the local healthcare delivery structure to address health and ill-health related issues from a holistic perspective through illness narratives of CKDu patients.

## 2. Materials and Methods

Our ethnographic study focused on lay perceptions of CKDu and healthcare seeking behaviour in two Divisional Secretariat (DS) divisions in Anuradhapura district in the North Central Province in Sri Lanka. As part of qualitative study on "social epidemiology of CKDu", the present paper analyses how the curing process moves between medicine and traditional culture. The transcendental phenomenological approach was used in our ethnographic study as it was focused on rather under-researched social phenomena. This approach seeks to describe the essence of a phenomenon by exploring it from the perspective of those who have experienced the same, noting what was experienced and how it was experienced (Moustakas 1994). The transcendental phenomenological approach holds the view that the reality is internal to the knower, what appears in their consciousness is bias-free and one must strive to understand the phenomenon by descriptive means (Neubauer et al. 2019, p. 92). We sought to learn through individuals' lived experience from their own perspectives. People make sense of well-being as it is experienced in their daily lives, and the transcendental approach enables us to describe how participants express the lived experience from their own perspective in relation to curing practices. We used a mixture of ethnographic methods throughout the data collection period. In data analysis, the phenomena were considered from different perspectives, with identified clusters organized into themes in order to form textual descriptions.

### 2.1. Setting and Subjects

Medawachchiya and Padaviya DS divisions in the North Central Province in Sri Lanka were selected for our ethnographic study. The North Central Province recorded the highest number of CKDu patients in Sri Lanka (De Alwis and Panawala 2019; Ranasinghe et al. 2019). Within the province, the above two DS divisions have been reported as "CKDu hotspots" due to the high prevalence of the disease since it was first identified as a health hazard in the dry zone areas in Sri Lanka in the 1990s (Ranasinghe et al. 2019; Jayatilake et al. 2013). These two settings seemed most suitable for our ethnographic inquiry to grasp the lay perspectives of CKDu that have been constructed and moulded through local discourse over the past three decades. Structural differences in these two settings were considered during selection as they offer a comparative perspective. Medawachchiya DS division is closer to a traditional village structure in Sri Lanka with strong kin-networks and cultural practices. By contrast, Padaviya DS division is predominantly an agriculture-based resettlement village established by the government under its resettlement programme, implemented in the 1950s. People from different parts of the country migrated into this area and settled down as farmers. The original migrant families received sufficient land for both paddy faming and high-land cultivation. However, the second and third generations of the settlement face a shortage of land as siblings of original migrant families have to share the land that they initially received.

Medawachchiya DS division consists of 37 Grama Niladari (GN) divisions. A GN division is the smallest administrative unit at the grass root level within a DS Division. Padaviya DS division consists of 15 GN divisions. We selected two GN divisions, one from each DS division, for our ethnographic study by considering the high prevalence of CKDu within the above two DS divisions. Identity of the selected two GN divisions in

our ethnographic study remains anonymous due to ethical considerations. A purposive sample of 60 participants who were diagnosed as CKDu patients and their households were included the study for qualitative interviews, with 30 from each setting. Snowball techniques were used to find the required number of patients from the two study settings. Initial contact with participant (CKDu patients) was through the two administrative officers known locally as "Grama Niladari". Other participants were gradually included the study through the social networks of initial contacts. We also included older adults who have life-long experience in their respective communities, leaders of community based organizations, various types of healthcare providers and social service officers in the two localities. Subsequently, ten households, five from each setting where a death was reported due to CKDu, were included the study to scrutinize insights of an entire tragedy from beginning to end.

### 2.2. Data Collection

Fieldwork in the two settings was carried out during a period of twelve months from 2017 to 2018 and continued with several follow-up visits until the end of April 2019. The researcher was on sabbatical to conduct fieldwork and the fieldwork schedule was prepared accordingly. Two research assistants (RA)—one for each setting—were recruited to assist the researcher during fieldwork. Both were graduates with relevant research experience and fluent in the local language. The two RAs were recruited to assist the researcher for notetaking when conducting in-depth interviews, focus group discussions and key informant interviews and to transcribe recorded interviews. The researcher had to recruit two RAs, one for each setting, as neither was available for prolonged periods of time during planned fieldwork. Our fieldwork started from Medawachchiya DS division in April 2017. The RA assigned to Medawachchiya DS division was from the same locality within close proximity—less than an hour by public transport to the field site. We began fieldwork in Padaviya DS division in early November 2017 immediately after conducting fieldwork at Medawachchiya DS division. The RA assigned to Padaviya DS division was from another region and resided in the field site during fieldwork. Regular meetings were conducted with the respective research assistants during the time of fieldwork in each setting. The researcher carried out several follow-up visits in the two settings simultaneously during weekends and in vacation times.

A mixed method approach was used in data collection and included participant observation, in-depth interviews, focus group discussions and key informant interviews. During fieldwork, the researcher took part in most of the situated activities in two communities in their natural settings including the harvesting ceremonies; annual celebrations such as Sinhala/Hindu New Year; weddings, funerals and regular meetings of community-based organizations including the Farmers' Organization, Funeral-aid Society, Women's Association and several religious and ritual practices. The researcher jotted down the main points gathered from participant observations on the spot. Most of the time, the detailed accounts in field notes were completed on the same day. In-depth interviews were carried out with participants (CKDu patients and/or caregivers of households) who were selected for the study by using a purposive sampling method. In-depth interviews were also conducted with family members of deceased CKDu patients. We conducted interviews with participants in their local language at their residential places and at times convenient to them to minimize the negative impact of recalling past events. We allowed them to speak up in their own way, using their own vocabulary, and gathered necessary information using a phenomenological perspective. This convenient approach helped them express their own interpretations regarding the illness experience in their everyday life. Interviews were recorded with the consent of participants and transcribed by the respective research assistant. Supplementary to the illness narratives, focus group discussions and twenty key informant interviews were conducted. Two focus group discussions (FGDs) constituting of a male and a female group were conducted in Medawachchiya DS division and included twenty total participants (8 in the male group and 12 in the female group). The two FGDs

at Padaviya DS division included twenty-two participants (12 in the male group and 10 in the female group). The two groups in each location were divided on gender basis to avoid power gaps in discussion. The historical background of the study setting and issues related to lifestyle changes, the process involved in identifying CKDu as a health hazard in concerned communities and their engagement in the process of determining etiology of the disease, illness management strategies, treatment options and cultural practices were discussed in detail while conducting FGDs with all four groups. Furthermore, in both settings, twenty key informant interviews were conducted. They included older adults (5) in respective settings, community leaders (4), different healthcare providers (9) and social services officers (2). All FGDs and key informant interviews were tape recorded and transcribed by the two research assistants.

Ethics clearance for the study was obtained from the Ethics Review Committee for Social Sciences and Humanities (reference No: ERCSSH/16/24) at the Faculty of Arts, University of Colombo, Sri Lanka.

### 2.3. Data Analysis

The data generated from participant observation, patients' narratives through in-depth interviews, FGDs and key informant interviews were analysed using the thematic analysis model proposed by Schweitzer (1998) and Giorgi (1994, 1997). The analytical procedure included stages of familiarization with the data, generating initial codes, searching for themes, revealing potential themes and defining and naming the themes. The whole process was manually accomplished by the researcher. All interviews were recorded and interviewee was given a pseudonym. Topics discussed were corded and organized into themes and sub-themes. During analysis, transcripts and observational notes were re-read repeatedly as a continuing process to identify patterns emerging from data. These are presented under the relevant themes in the form of direct quotes, narrations and summary statements to explore lay perceptions of CKDu and to describe individual's own perspectives of curing between medicine and cultural practices. The direct quotes of participants are indicated in italic and location of participants remains anonymous to minimize negative consequences. However, location of FGDs can be identified in direct quotes as context specific issues need to be highlighted.

### 3. Results

Patient narratives reveal that the majority of patients in both Medawachchiya and Padaviya DS divisions were diagnosed with CKDu during the productive age in their life-span. The majority of patients are male farmers functioning as the head of household as well as the main breadwinner of the family. Patients' narratives further confirmed that the majority were poor even prior to diagnosis as livelihoods are agricultural with uncertain and low incomes. Previous studies elaborate on the diverse economic and psycho-social problems faced by affected communities in addition to loss of productivity due to illness and the costs of caring for patients (Liyanage 2019; WHO (World Health Organization) 2016). The results of the present ethnographic study are organized under five themes: 1. Identifying CKDu as a health hazard and discovering unknown etiology by experts; 2. CKDu discourses among affected communities; 3. categorization of illness through narratives; 4. local dynamics of healthcare seeking behaviour and a demand for complementary medicine; 5. drawbacks in the healthcare delivery system and a transformation of CKDu hotspots into sites of humanitarian concerns.

### 3.1. Identifying CKDu as a Health Hazard in the Local Context

CKDu was initially identified as a health hazard in both Medawachchiya and Padaviya DS divisions in 1990s by chance when patients visited local healthcare institutions seeking treatment for symptoms such as back pain, muscle pain, swollen body, urine infections and continuing fever, etc. (Abeysekera et al. 1996; Athuraliya et al. 2011; Jayatilake et al. 2013). By scrutinizing those symptoms, local healthcare professionals were able to diagnose

the above symptoms as being associated with chronic kidney disease. After thorough investigation through a medical gaze, healthcare professionals confirmed that the etiology of the newly detected disease was not similar to conventional explanations with regard to chronic kidney disease (Abeysekera et al. 1996; Athuraliya et al. 2011; Jayatilake et al. 2013).

In response to this emerging health hazard, the biomedical system played a dominant role. Priority was given to providing healthcare services for patients while conducting diverse research to determine the unknown etiology of the disease. Parallel to this process, massive screening programmes were conducted at grass root levels to identify susceptible patients in endemic areas (De Alwis and Panawala 2019; Jayatilake et al. 2013). Medawachchiya and Padaviya DS divisions have been identified as high prevalence areas and labelled as "CKDu hotspots" since the beginning. The identified patients were categorized into five groups by biomedical practitioners based on criterion assessing the severity of the disease. Clinics were setup in endemic areas to provide treatment for patients. The patients who were in the early stages of disease were instructed to attend the clinic once a year for follow-up. Those in the middle stage of disease progression were referred the clinics once every six months for follow-up and those who were at critical stages were instructed to attend regular monthly clinics set up in their locality. The system initiated during the early phase of the endemic has continued without drastic changes. Patients' narratives reveal that there are a number of gaps in service delivery mainly due to different perspectives and distant relationships between the biomedical healthcare providers and CKDu patients. As revealed through all our primary data sources, CKDu articulates in affected communities as "a struggle between life and death" where the well-being of patients, family members and the entire community has deteriorated. Empirical evidence reveals that CKDu patients experience diverse psychological disorders such as fear, anxiety, anger and sleeping disorders, particularly those who live in a place labelled a "CKDu hotspot" where causative factors seem uncertain and available options for curing are minimal. The patients and their families did not receive sufficient information from the clinic to manage ill-health related issues. The current system neglects to integrate these crucial psycho-social support services into the clinical setting.

As indicated earlier, over the past few decades a large number of studies were undertaken to determine the unknown/uncertain etiology of this newly identified disease by researchers from diverse disciplines. Biomedical research on CKDu included population prevalence and case-control studies; metal analysis of urine; analysis of hair and nail samples of patients for arsenic; renal biopsy studies and post-mortem specimens of CKDu patients such as kidneys, livers and bone tissues (De Alwis and Panawala 2019; WHO (World Health Organization) 2016; Jayatilake et al. 2013). The environmental studies focused on testing samples such as irrigation water, agro-well water, soils of agricultural and non-agricultural lands, rice, vegetable, freshwater fish and other food items and nephrotoxic herbal remedies from high CKDu prevalence areas (De Alwis and Panawala 2019; Jayasumana et al. 2016; WHO (World Health Organization) 2016). However, findings from these diverse studies resulted in only a large number of hypotheses without solid conclusions on the etiology of the disease (Rajapakse et al. 2016). Furthermore, various studies carried out in other CKDu affected regions further confirmed the absence of a solid conclusion with regard to etiology of CKDu, though suspected causes were highlighted in their studies (John et al. 2021; Paidi et al. 2021; Lunyera et al. 2016).

Four FGDs in the two settings clearly suggest that lay people in affected communities were marginalized throughout the investigation process to determine the unknown etiology. Their involvement was minimized to merely providing objects (hair, nail, kidney tissue, water sample, etc.) required for scientific investigation by biomedical and environmental scientists. The biomedical explanation of disease focuses predominantly on biological changes to the relative neglect of social, cultural and psychological factors. It is clear that the lay perception of the disease is totally neglected throughout the process of discovering etiology. Subjective interpretations were deemed irrelevant and patients were considered as passive objects, their involvement limited to only providing necessary

objects. FGDs in the two settings reveal that these communities attracted a large number of researchers mainly due to high prevalence rates of CKDu. The environmental studies also operated a top-to-bottom approach by experts from various disciplines with contradictory conclusions, creating further confusion among affected communities (FGD No 1 and 2 in Medawachchiya, and FGD No. 3 and 4 in Padaviya). There are no remarkable contextual differences between the two settings with regard to identifying CKDu as a health hazard in two study settings as a similar processes operated in "CKDu hotspots" to identify patients, discover etiology and provide treatments.

### 3.2. CKDu Discourse among Affected Communities with Regard to Uncertain Etiology

Similar to expert groups, lay people also have their own discourse with regard to CKDu. This has been constructed and developed through the lived experience of participants, explanations given by experts who investigated the unknown etiology of the novel disease and beliefs in traditional culture. This discourse explores lay-perceptions and interpretations regarding the origin, etiology and prevalence of CKDu in their locality during the past few decades.

It is a dilemma within the two communities whether CKDu is a recent phenomenon in their localities or if it prevailed in the past. Most of the patients believe that the tragedy of CKDu is a recent phenomenon in their locality due to adverse effects of socio-economic and cultural transformations that occurred over the past few decades. However, recalling past experience, a few older adults who participated in a FGD at Medawachchiya pointed out that they had witnessed a similar type of illness with symptoms such as a disfigured-face, swollen body and breathing difficulties that was known locally as "*pittapanduwa*" or "*pipihaluwa*" (anemia or a condition of malnutrition), though it was not diagnosed as kidney disease. They believed that those symptoms were caused by a lack of blood in the body of the patient, and therefore he/she was treated with foods which villagers considered as the most nutritious and included various types of meats (*Dadamas*), milk, curd and certain types of locally available leaves (FGD No 1: male group at Medawachchiya). Adding to the discussion, one of the older adult participants noted that he was aware of some incidents where a number of patients presented with similar symptoms, compelling other villagers to migrate to another setting, believing that their locality was under the influence of demonic attack (FGD No 1: male group at Medawachchiya).

When I inquired about similar experiences from participants of the two FGDs at Padaviya, they were not aware of it. However, during a key informant interview with an older adult participant at Padaviya, he referred to his childhood experience as a member of an original migrant family. He was aware of a condition of "*pipihaluwa*" prevailing in dry zone areas, though he never personally witnessed such cases in his lifetime at the resettlement village. In contrast to the preceding view, a vast majority of villagers in both settings strongly believe that CKDu is clearly a recent phenomenon resulting from the mismanagement between the natural environment and the lifestyle changes of residents during the past few decades. Therefore, they view CKDu as a man-made disaster. The previous generations never experienced such a situation as they were capable enough to manage a harmonic balance between the natural environment and their social environment (FGDs 1 and 2 at Medawachchiya and FGDs 3 and 4 at Padaviya).

The four FGDs in Medawachchiya and Padaviya revealed how their water sources became polluted with the heavy usage of pesticides and chemical fertilizers in paddy and high-land cultivation. The tank was the main source of water for many purposes, and it was organized as a cascade system. There were distinct spaces allocated in it for each activity and the villagers had their own community-based systems like "*shramadhaana*" (voluntary participation to fulfil a particular task collectively free of charge) to maintain the village tank regularly. Diverse religious and ritual practices also contributed to protecting and maintaining the purification of the tank. The participants of FGDs explained how the village tank and other water sources became polluted due to changing agricultural practices, notably the heavy usage of pesticides and chemical fertilizers.

Key informant interviews with older adults and the information shared by participants at four FGDs in both settings clearly demonstrated that healthy, eco-friendly and sustainable traditional agricultural practices existed in their localities. The staple food of rice was cultivated in paddy fields while vegetables, greens, grains and cereals were cultivated in rain-fed lands commonly known as "*chena cultivation*". The *chena cultivation* was also known as shifting or slash-and burn cultivation in which villagers searched for abundant land and prepared it for cultivation before the rainy season started. Immediately after the initial rainfall, they cultivated a huge variety of food items in *chena* which were completely organic and enabled them to consume a balanced diet to maintain a healthy life. The *chena cultivation* that provided diverse food varieties was gradually disappearing due to land shortages, risk of elephant attacks, etc. In response to the disappearing *chena cultivation*, villagers have started cultivating these food varieties in their home gardens which were previously for cultivating herbal medicine for minor ailments and supplementary cuisine flavourings found very often in cooking. Changing patterns of cultivation from *chena* to the small home garden plots demand heavy usage of chemical fertilizers and pesticides to cultivate similar type of crops repeatedly on the same land.

As one of the participants pointed out:

*"One day I sprayed some fertilizers into my home garden and in the evening there was rain. Next morning when I went to the well, I saw hundreds of frogs died near the well. I am sure this was because of the fertilizer that I sprayed. This is the well that we use to get drinking water. So we can understand the bad effect of fertilizers, but we are helpless as there are no alternatives for this problem"* (CKDu patient who cultivates in home garden).

The popular discourse of CKDu among affected communities reveals that there are many causes of CKDu, including unhygienic drinking water, heavy usage of pesticides and chemical fertilizers, dehydration due to hot weather in the dry zone, snake bites, genetic factors, hypertension, diabetes, heavy use of alcohol, long term usage of medicines and hard work associated with their livelihood activities. Even though the participants have identified many possible causes for CKDu, they consider the heavy usage of agrochemicals and the poor quality of locally available drinking water as the main causes of CKDu. Therefore, they are in search of potable water from suitable sources even bearing out-of-pocket expenditure. An older adult male participant admitted that excessive usage of agrochemicals has polluted the area heavily and it is worse than it ever has been. As he pointed out,

*"Not like in earlier times, all the farmers here are now depending heavily on agrochemicals. Water which flowed from culverts used to be exceptionally clean and drinkable and full of natural aquatic organisms during rainy seasons. Now, however, the water is inherently polluted"* (An older adult male CKDu patient who was a farmer).

Overall, we found that participants were largely unsure about what causes this illness. However, when asked why they thought the two main causes of CKDu were the heavy usage of agrochemicals or poor quality drinking water, most respondents mentioned that agrochemicals could have drained into their wells or there could be something harmful in the drinking water they have been consuming. People implicated poor quality drinking water as the main cause for CKDu. Some participants at the female FGD in Padaviya pointed out that:

*"We cannot use the aluminum pot that we use for water storage for not more than a year as it gets damaged due to chemicals in the ground water. This is a good example for you to understand the damage that can happen to the kidneys of human beings"* (FGD No. 4: Female group at Padaviya).

This has led some villagers to walk several miles to obtain daily supplies of water from sources they believe are not contaminated (from spring wells, bottled water, water filters etc.), adding additional costs to the household budget. All patients in both settings

have changed their drinking water source from ground wells to the Reverse Osmosis (RO) system. There is a community-based system to operate and maintain the RO watershed in the village that subsidizes families of CKDu patients.

Snake bite was also indicated as one of the risk factors associated with CKDu in Sri Lanka and in other regions (Lunyera et al. 2016; De Alwis and Panawala 2019). This hypothesis was developed after investigating case histories of CKDu patients and observing that a large number of patients have experienced a snake-bite attack at least once in their life prior to diagnose as CKDu patients (WHO (World Health Organization) 2016). Gender difference in the rates of CKDu indicate a trend of male prevalence where two out of three CKDu patients are men (Ranasinghe et al. 2019). While reviewing lived experience of CKDu patients in their local context, the participants of this study remark that snake-bite attack could be one of the main risk factors for this pattern of CKDu. As they explained, men are comparatively more vulnerable than women to be affected by snake bites and the venom could be damaging their kidneys. Adding to the discussion, they pointed out that a very dangerous snake unique to the dry-zone called "*gata-polangga*" could be easily found in their community (FGD No 3: male group at Padaviay). These participants believe that snake-bite attack could be one of the main reasons for higher reported male CKDu patients in their locality. Additionally, men are more vulnerable than women to snake-bite attacks as they go out at night and spend more time during the day in their farmlands engaged in agricultural activities. Initially, they gathered this idea from experts who attempted to explore the unknown etiology of novel kidney disease and connected it with their lived experience. The participants of both male and female FGDs at Medawachchiya also believe that snake-bite attack could be one of the causative factors of CKDu as a vast majority of patients have a history of snake-bite attack in their locality. Participants of the female group at Padaviya accepted snake bite as a risk factor and added another explanation connected to the behavioural patterns of men. Specially they noted that most of them consume illegal liquor produced in their locality and have a habit of smoking (FGD No 4: female group at Padaviya).

Genetic factors are also seen as one of the causative agents, as in the same family a number of members suffer from CKDu. As is evident from our ethnographic investigation, we found ten families in Padaviya and three families in Madawachciya having two/three patients. As one of the patients pointed out:

> "*I am sure I got this problem from my family. My grandfather died because of kidney problems and then my father and his elder brother too died from kidney disease. So I hope there is more than enough evidence to support this argument*" (Male CKDu patient with a family history).

The perception that CKDu can be transferred from one generation to another has negative social consequences as a certain amount of stigma is attached with the disease. This is highlighted by villagers' experiences and their voices. Lay persons, similar to professionals, are unsure as some of their experiences go against rational explanations on the causes for CKDu. As a result, they tend to justify and receive some consolation by attributing it to "Karma" (bad luck). As narrated by a patient:

> "*I know of a person in our village who never drinks boiled or/and filtered water. He engages in wage labour and spray fertilizes. He takes alcohol daily. However he is ok, but I am sick even though I am careful in my behavior and take precautions to prevent illness. So I think this is my bad luck/karma*" (Male CKDu patient).

It is significant to note that lay people, similar to experts, continue the discourse on the etiology of CKDu by further analysing their illness experience.

### 3.3. Illness as Perceived by Subjects

Illness narratives of CKDu patients and data gathered from four FGDs at Medawachchiya and Padaiya DS divisions reveal a similar pattern of illness categorization that includes three types of kidney diseases commonly known as: '*vakugadu idimeema*' (kidney swellings); "*vaku-*

*gadu diyaweema*" (kidney dissolving) and *vakugadu ekileema* (shrinking of kidneys). Initially, patients gathered these terms from the hospital. However, peasant society interpreted them by integrating observed day-to-day phenomenon in their surroundings related to their agricultural livelihood. They were not able to explain fully what these terms really meant but simply repeated that doctors used them. Being farmers for years and having gathered the experience from nature and its calamities, it seems people have feared these terms without proper information. Patients from both settings commonly utilized tertiary care services, and the same nephrologists served in CKDu clinics in both settings. It could be the reason for the construction of similar categories of illness by lay people in those two settings.

Interestingly, "shrinking" equates to the process that occurs during the wilting of maize. They assumed that the dissolution of kidneys is similar to the process of dissolving "urea", a chemical fertilizer commonly applied by many farmers to enhance soil fertility. Contrary to the biomedical explanatory model, the shrinking of kidneys is not considered kidney disease within village discourse. According to their narratives, the physical appearance of people who are suffering from "shrinking of kidneys" looks very healthy as there are no visible symptoms except the protein that passes with urine, something they came to know from screening programmes conducted in their locality. According to their experience, urine infections are common in their locality due to the dry climatic conditions. They can be prevented by eating different varieties of cucumber and by herbal medicines which are commonly available in their locality. They believe that the cause of this type of illness is due to a lack of water intake in daily life. As narrated by a patient:

> "*I was identified as a CKDu patient at the mobile clinic in 2005. My urine sample was not good at that time and I was asked to repeat the test for another two times and found that protein passes with my urine. Therefore, I was referred to Kandy hospital for further checkups where I lost a part of my kidneys as doctors wanted a piece of my kidneys for their investigations. However, still I do not know the outcome of that test. I was referred back to the nearest hospital for treatment where I was asked to come for checkup once in 6 months and later on once a year. Last month only I attended the clinic and doctors said that I am ok. I had only a urine problem but not the CKDu. I drink polpala or ranawara (herbal medicine) for my urine problem as advised by my mother. After I started taking herbal medicine my urine problem has disappeared. So, I am ok now. However, I am bit worried as I have lost a part of my kidney. Will it be a problem for my kidney?*" (Female CKDu patient).

The dissolution of kidneys is connected with the damage caused by the impact of poisons entering inside the body. As a result, the kidneys are gradually dissolving and not functioning properly. The common symptoms are breathing difficulties, back pain and headache. They believe that the cause of the disease is mainly due to the impact of venom that entered the body through heavy usage of chemical fertilizers and pesticides applied during the cultivation of rice and vegetables, unhygienic water for drinking, snake bite, etc. They also believe that it would be impossible to recover the damaged kidney, but it could be controlled by taking medicine. During our interviews with CKDu patients, a number of them volunteered to share their experiences regarding doctor–patient interactions in their therapeutic encounter. Describing her illness experience, one of the patients pointed out that:

> "*The doctor said to me at once that you have the kidney problem that is spreading around Rajarata. The doctor showed me the image on TV screen and he told –look at your kidneys, it is small, and it is dissolving, only 10–15% of your kidneys are working. I couldn't eat for weeks after hearing about it, I believe that the doctor should not say it to our face like this*" (Female CKDu patient).

Another patient sharing his experience noted that:

*"Some of my friends were told (in the clinic) that they have the problem of shrinking kidneys, but I was never told what I am having. One day I inquired about this from the doctor at clinic. He kept quiet without answering"* (Male CKDu patient).

A further patient added

*"I was screened for CKDu, when I got severe back pain –they said my kidneys were dissolving"* (Older adult male CKDu patient).

The illness narratives above reveal that there are a number of gaps in doctor–patient interactions where patients are not passive but attempt to explore the meaning of illness through their lived experience and associate signs and symptoms with those illness categories. Accordingly, some of the patients believe that shrinking kidneys are mainly due to a lack of daily water intake. Most of the patients relate their problem either with dissolving kidneys or swelling kidneys. According to their interpretations, these two types of kidney diseases are mainly due to the absorption of chemicals into the human body (*angata wasa visa athuluweema*) through contaminated water, food produced with intensive use of chemical fertilizers/pesticides and snake bites that are a common occurrence in their locality.

Most of the patients consider that the swelling of kidneys is comparatively the worst category of CKDu. It leads to severe breathing difficulties, swelling of the whole body, disfigurement of the face and loss of appetite. The patients who are recommended for dialysis are simply placed into either dissolving or swelling of kidneys by the concerned community. Though they have identified chronic back pain, hypertension and generalized body edema as the main symptoms of the illness, some people have ignored their pain, assuming it was probably due to excessive labour as farmers do work quite long hours in their paddy fields.

Once the kidneys are affected, they all know that the ability of the body to purify the blood was altered. Patients who have undergone dialysis or are referred to have dialysis know they had reached the end stage of the disease despite insufficient knowledge of the procedure of dialysis and its outcome. It was noticed that relatively young, educated patients gained the knowledge to understand the results of their blood reports. However, it was found that they still had not gained the knowledge of disease progression and available treatment options. The lack of knowledge could be the main reason for refusing to undergo dialysis whenever necessary.

*3.4. Local Dynamics of Healthcare Seeking Behaviour and Emerging Demand for Complementary Medicine*

It was found that all patients had been registered at the nearest biomedical clinic established for CKDu patients. Every patient acknowledged that they were diagnosed clinically by biomedical clinics in the area. Only a few patients visited the private hospitals in urban areas to get confirmation of the diagnosis made at the clinic. The illness narratives of participants revealed that they had little knowledge regarding treatment procedures at the CKDu clinic but rather were confused as all of them were given a similar type of medication irrespective of the different categories of illness explained earlier. Based on their three decades of experience, the community had concluded CKDu was an incurable deadly disease. They were aware that the medication given by the clinic only slowed the progression of the disease. The situation had become rather complicated due to the different opinions given by medical practitioners in healthcare management centres. As narrated by one of the participants:

*"I was told by the doctor at the clinic that my kidneys are dissolving, but we had a consultant who came from outside to examine us from time to time. So, I asked him of what type of CKDu I'm having now. He said (the visiting consultant) there is no such a thing called kidney dissolving—so, I am confused. The consultant must be correct because he is an expert who came from a big hospital"* (Male CKDu patient).

All the patients maintained that they were not properly educated about the illness by the physicians. They have come to the conclusion that medical officers in the clinic have no sufficient knowledge about the disease as they are not aware of the causes/etiology of the disease. As a group of participants highlighted in a focus group discussion at Padaviya:

*"What we know is what they (doctors) know, so what is the point of asking about the illness from them if they have no adequate knowledge about the disease. In fact, they didn't tell us anything when we asked about our present situation. All they say is to take medication. But we know the medication does not cure the illness"* (FGD 3: male group at Padaviya).

The patients also pointed out that the quality of care given in the hospital was once quite high but has diminished over time. As one of the patients commented:

*"At the beginning they (the physicians) recommended more blood tests per year and also they measured our weight, during examination. Now they don't measure our weight and now we have to ask for a receipt to check our blood, they only measure blood pressure."* (Male CKDu patient).

The participants pointed out that their quality of life never improved, even though they had been taking medication for months or sometimes years. Most of the patients added that they had never experienced any physical comfort. This is most likely why patients in the area have selected Ayurveda/traditional medication to replace Western medicine. Thus, we observed that the number of patients who seek Ayurveda medication increased. We interviewed a group of patients who had just registered in the nephrology clinic but did not adhere to any advice or medication. As one of the participants pointed out:

*"We attend the clinic, only to take the 5000 rupees, not for any other reason. We take the medication and throw them away or give it to others who take western medication regularly. There are occasions where some medications not available at the hospital and patients are requested to buy them from the pharmacy. So, I usually store the medication to share with those who need them"* (Male CKDu patient).

As mentioned earlier, the government initiated a monthly cash subsidy for CKDu patients to support them. The patients who utilized Ayurveda simultaneously attended the CKDu clinic just to maintain a good record of attendance as it is evidence that they fulfilled the eligibility criteria for the monthly allowance and it preserved the choice to utilize the hospital in case of an emergency. Most of those patients pointed out that they had a guilty feeling as they discarded the medications given by the government hospital free of charge and used Ayurveda medicine, paying out-of-pocket. However, the patients rarely shared their experiences of Ayurveda/traditional medicine for CKDu with healthcare providers while attending the regular clinic, assuming the negative consequences they might receive in return. As one of the participants noted:

*"The western doctors never encouraged patients to use Ayurveda/Traditional medicine though it gives us relief. If they come to know that we are using Ayurveda they might get annoyed. So, it would be better to be silent to avoid unnecessary problems"* (Male patient recommended for dialysis).

Patients' narratives suggest that there is a huge gap in doctor–patient interactions. However, while patients seemed very silent in the clinical setting, they were very dynamic in other domains of their lives. They sought Ayurveda/traditional medicine from the practitioners who practiced within the locality and the more distant practitioners. Social networks and lay referrals played a crucial role in searching for multiple therapeutic options. In comparison with Western medicine, patients highlighted some of the improvements in their health condition with the commencement of Ayurveda/traditional medicine. According to the experience of those patients who utilized Ayurveda medication regularly, the functional capacity in their everyday life has improved to a great extent and they are involved in former pursuits again. Thus, reduced sleeping disorders, improved appetite and an added a hope for life were mentioned. Referring to the few failure cases in their

surroundings, patients blamed those individuals, not the system, claiming they did not adhere to given instructions and medicine properly to manage the illness. They reiterated the fame of historical accounts of Ayurveda medication which were used in Sri Lanka for more than 2500 years before the introduction of western medication. On questioning, they emphasized that Ayurveda medications are made by the plant products known to them. They also said that the relative quantities of components of decoction differed from patient to patient. As a participant pointed out:

> *"My friend and I have different types of kidney problems. So, we were given different medication by the Ayurveda practitioner. But in CKDu clinic we both are given all the same medication."* (Male patient who utilized Ayurveda for few months).

As previously stated, the villagers feared undergoing dialysis as they were under the assumption that it meant death was imminent.

> *"They (Doctors) asked me to undergo hand preparation procedure. I know once it was done, I can't continue my usual activities. I know all the people who had dialysis, they died within a very short duration"* (Male patient who used Ayurveda after being recommended for dialysis).

It was noticed that most of the patients who were recommend for dialysis felt compelled to seek alternative treatment methods. We met five patients who refused to undergo dialysis. All of them mentioned that they did not want to insert any instrument into their body because if so, they would have to live with that instrument until their death. On further questioning of dialysis, they pointed out it was a lengthy, painful procedure and if the doctors find that their arm is not suitable–their next target is the neck. All these patients pointed out that dialysis brings patients to their death faster, recalling all of the names of neighbours/relatives who have died in the past, even though it seemed that they tried their best to avoid undergoing dialysis. This kind of attitude compelled patients to turn to an alternative medication where there were no known invasive procedures available. Patients who underwent dialysis showed us their disfigured hands. As one of them stated:

> *"It never ends, there will always be more dialysis you need to have done in future."* (Male patient who had undergone the process of dialysis).

Another participant pointed out:

> *"Every patient who has had to undergo dialysis has quit their job and now lives an uncomfortable, unfulfilling life"* (Male CKDu patient).

Narratives reveal that most of the patients did not trust the medical report given by the clinic, so they habitually repeated the blood test in private centres at Madawachchiya or Padaviya. Some of them used to take tests from centres further away in Anuradapura, Kandy and Trincomalee. They gathered these reports and compared the values of one over the other. All maintained that their final referral was to the government hospital in an emergency, and therefore they tried their best to maintain a superficially good relationship with hospital staff. They pretended to follow the exact instructions given by the doctors, fearing they would not receive further treatment if the hospital staff realized that they had ignored the instruction given to them. As one of the participants remarked:

> *"Even though we don't take medicine, we attended the clinic regularly, so on emergency we have proven that we were very obedient to the doctors' instructions, by showing our clinic record books"* (Male patient who used Ayurveda after being recommended for dialysis).

Patients were assured by Ayurveda medical practitioners that CKDu is treatable and has a permanent cure, provided the patients follow instructions assiduously. Patients were comfortable with the ingredients used in preparing medicine because they were known to them and were in abundance around the area. Some of the ingredients have been known for the treatment of urinary tract infections and therefore patients were aware of them and were knowledgeable regarding the uses and effectiveness of that medication. During our

interviews, some patients claimed that they were completely healed. Patients believed they could avoid dialysis if they started complimentary medication soon after the diagnosis was made.

One thing that all participants accepted as true, which is not the case in reality, was that once a patient was told to undergo dialysis death was imminent. Consequently, they tried their best to avoid dialysis. It was not known by patients that dialysis was their final option when they enter the last stage. Patients were asked to come and prepare their arm for dialysis. Most of the male patients were the breadwinner of the family and were not able to totally step away from their responsibilities and accept their sick role. Five of the patients who were recommended for dialysis have moved to alternative medicine. However, within the two communities there was a discourse regarding the role of complementary medicine for CKDu, and some patients were reluctant to move from western medicine to Ayurveda/traditional medicine. All the patients adhered to meditation and religious healing techniques as complementary to Western or Ayurveda medical systems.

The lay explanatory model of CKDu and the healthcare seeking behaviour of patients revealed that Ayurveda/traditional medicine played a key role as a complementary system which helped patients to overcome some of the limitations of the dominant system of biomedicine. Illness responses of CKDu patients support the argument of a non-system approach where health seeking behaviour was predominantly determined by pragmatic factors such as the comfort and improvements of daily functional capacity of patients, friendly healer-patient relationships and positive feedback with an increased hope for life, etc. (Liyanage and Ekanayaka 2018; Silva et al. 1994; Caldwell et al. 1989). Lay referrals of patients played an important role in both decision making and in finding suitable practitioners even from distant places. Family bonds of patients were further strengthened through the healing process as it required a collective effort in preparing decoctions, applying medicinal oil, etc. Thus, Ayurveda/traditional system functioned as complementary where the healers encouraged patients to attend regular clinics and even to use their clinical reports for monitoring purposes. However, patients remained quite reluctant to share the experience of complementary medicine with western medical practitioners, assuming possible negative feedback.

There is an increasing demand for complementary medicine over the past few decades, particularly with regard to chronic health problems (Zörgő et al. 2018; Eaves et al. 2015; Ritenbaugh et al. 2011; Cant and Sharma 1999). A qualitative study on culturally embedded factors in complementary/alternative medicine use advocates avoiding a dichotomization between "push and full factors" and highlights significance of taking context specific factors into account (Zörgő et al. 2018). Some studies have focused on integration between complementary/alternative medicine and the "orthodox" medicine that goes beyond both dualism and pluralism (Wiese and Oster 2010). Integration involves the selective incorporation of aspects of complementary/alternative medicine to be used alongside biomedical treatment (Wiese and Oster 2010). This integration involves respect and collaboration between different views of health and healing and requires a mutual transformation that may not always be acknowledged by different types of healthcare providers (Glasby 2005). Findings from our ethnographic study suggest that working relationships between modern and traditional practitioners is essential to strengthen responses and deal with contemporary health hazards in a context where dissatisfaction with the healthcare outcomes abounds. It is obvious that Ayurveda/traditional medicine offers a complementary approach that enables participants to minimize some of the negligent features in the dominant healthcare delivery system. However, the provider perspective requires more collaborative efforts to ensure a complementary system given the strong desire from the user perspective.

*3.5. Drawback to the Dominant Healthcare System and the Transformation of CKDu Hotspots into Sites of Humanitarian Concern*

CKDu is a novel experience for both the patients and the healthcare providers. Our ethnographic study examined insights of the popular explanatory models of CKDu which have been constructed through the lived experience of laypeople during the past three decades. The popular explanatory model of CKDu highlights a number of gaps in interventions addressing the emerging health hazard with an uncertain etiology. The findings indicate that the affected communities have been marginalized in the process of discovering unknown/uncertain etiology and in the decision-making process of ill-health management. The biomedical system itself has a number of limitations including: championing mind-body dualism; explaining disease by predominantly focusing on biological changes to the relative neglect of social, cultural and psychological factors that are also associated with the problem; and assuming every disease is caused by a specific identifiable agent/disease entity (Nettleton 1995). The illness narratives of CKDu patients revealed that enormous discrepancies exist in doctor–patient interactions; the doctor attempts to configure the disease through the medical gaze and adopts a rather dehumanizing stance, resulting in the segregation of a patient's body from the person. Thus, patients are treated as passive beings, their subjective awareness deemed irrelevant, and treated as a homogeneous category labelled as "CKDu patients" rather than acknowledging their human diversity. The situation causes further confusion among laypeople due to the medical language used in CKDu clinics by healthcare providers. Huge discrepancies occur in doctor–patient interactions which are not simply due to the disease's uncertain etiology but stem from a lack of required cultural competency, communication skills and humanities among the healthcare providers who practice in the areas where the disease has become a tragedy. Hence, in a context that is highly dependent on laboratory medicine, both the doctor and the patient are displaced by the scientific tests carried out in these laboratories. The experience of CKDu patients illuminates a number of limitations in the biomedical system dealing with chronic health problems which require a holistic and collaborative approach by including patients and healers in the context of culture (Kleinman 1980, 1988). In every culture, illness, the responses to it, individuals experiencing and treating it and the social institutions relating to it are all systematically connected. The health care system, like other cultural systems, integrates the health-related components of society (Kleinman 1980).

The specific context strongly demands an integrated and collaborative approach to understand CKDu as a community problem. However, most of the medical research was carried out in hospitals and the link between community and hospital in Sri Lanka is almost absent in CKDu research (de Silva 2015). The unhealthy compartmentalization has prevented a comprehensive understanding CKDu as a problem in the community rather than only at the level of individual patients (de Silva 2015). Findings of the present ethnographic study indicate an increasing trend of psycho-social issues in community settings not simply due to the uncertain etiology of the disease but also due to medicalized understandings of the issues surrounding CKDu which highlights the need to design interventions accordingly. Healthcare providers have not been able to convince the public that early diagnosis and treatment can relieve most of the problems associated with the later stages of the disease (de Silva 2015). When dealing with a community-wide disease like CKDu, it is imperative that a strong link between communities and the medical profession is forged in such a way that patients and families have confidence in hospital staff. Biomedical research on this complex syndrome has revealed little over the last few decades (de Silva 2015).

The evidence also indicates a necessary transformation from "CKDu hotspots" to sites of humanitarian concerns. Limited social protection mechanisms are in place to provide a minimum safety net for most vulnerable families, but they are not sufficient to fulfil the growing demands from the tragedy. Civil society organizations are also involved in providing social service for vulnerable sections in CKDu affected areas, including material benefits for children in education, drinking water facilities and logistics for patients who

come to tertiary care services, etc. However, the experience of participants reveals that there are some issues as most of those support services are irregular and also lack appropriate mechanisms for the fair distribution of services. The healthcare delivery system plays a dominant role in maintaining CKDu clinics in affected areas. However, patients have to spend out-of-pocket expenditure for various reasons.

Emerging psycho-social issues due to CKDu are rather neglected (Liyanage 2019). Considering its high prevalence, some of the areas have been labelled as "CKDu hotspots" to prioritize healthcare needs for the affected people in those areas. Unfortunately, labelling them has a social cost. The participants pointed out that they are experiencing several negative consequences due to identifying their localities as "CKDu hotspots". As mentioned earlier, CKDu is associated with sigma. We observed some cases where CKDu patients face difficulties in finding partners for their children as genetic factors have been indicated as one of the causative agents. During our fieldwork, we came across some incidences where patients were attempting to hide the illness to avoid negative consequences. Some participants pointed out that they feel embarrassed when others see them as an abundance of sympathy harms their dignity. However, further studies are required to examine how stigma interacts with CKDu and its implications on affected communities. Insights from lay perspectives of CKDu indicate these negligent features and create a demand for humanitarian concerns through an integrated approach.

## 4. Conclusions

Insights offered from the perspective of "lay" community members suggest the need for a conceptual shift within the ethno-medical model to address CKDu as a community problem. The conceptual framework needs to shift from individual to group, single explanatory cause to multiple or chain of causes and from curative to analytical approaches where the affected communities should be at the centre. The ethno-medical model with a holistic perspective enhances collaboration among multidisciplinary groups to bridge the gap between the hospital and the community. Healthcare professionals have largely ignored the socio-cultural context where they practice. The affected communities have been marginalized in investigations of the etiology of CKDu and in designing interventions. The affected communities have constructed a discourse on CKDu based on decades of their lived experience of the novel disease. Lay interpretations of CKDu highlight several gaps in knowledge sharing leading to misconceptions and unrest. Communication gaps in doctor–patient interactions, lack of cultural competency of healthcare providers and an absence of systematic psychosocial support services for patients/family members are highlighted through patients' journeys. The biomedical system plays a dominant role in providing healthcare services. However, a conceptual shift is required from biomedical to bio-psychosocial perspectives to address gaps in service delivery. Emerging discourses on the efficacy of complementary medicine demands an integrated system for healthcare delivery that requires policy alterations. A transformation is necessary from "CKDu hotspots" into sites of humanitarian concerns with multidisciplinary work that requires the inclusion of social workers into the healthcare delivery system. Improving culture competency and communication skills among healthcare providers in public health are crucial to accommodate a "bio-psychosocial perspective" in the healthcare delivery system in Sri Lanka. It is essential to continue the complementary discourse and explore strategies for integration. Further research requires identification of specific areas which can contribute to an integrated system.

**Funding:** This research received a Research Grant from the University of Colombo under its Small Research Grants (Ref. No. AP/3/2016/SG/05).

**Institutional Review Board Statement:** The study was conducted in accordance with the Declaration of Helsinki, and approved by the Ethics Review Committee for Social Sciences and Humanities at the Faculty of Arts, University of Colombo (ERCSSH/16/24) on 7 September 2016.

**Informed Consent Statement:** Informed consent was obtained from all subjects involved in this study.

**Data Availability Statement:** The data presented in this study are available on request from the corresponding author. The data are not publicly available due to privacy issues.

**Acknowledgments:** The author would like to thank all participants for providing their time and sharing lived experience of CKDu. The author acknowledges assistance received from Nishantha Wickramanayake and Sumithra Rajapakse as research assistants during fieldwork, Brianne Wenning for proofreading the manuscript and my loving son Thiran Liyanage for technical assistance.

**Conflicts of Interest:** The author declares no conflict of interest.

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
