# Peer review of "Chronic Kidney Disease of Uncertain Etiology in Sri Lanka: Curing between Medicine and Traditional Culture"

_socsci, doi:10.3390/socsci11010020_

Round 1

Reviewer 1 Report

General

This article focuses on the lived experiences of those suffering from Chronic Kidney Disease of unknown origin (CKDu) in two endemic ‘hot spots’ in Sri Lanka. Ethnographic methods were used to explore local communities’ own perception of the disease and on the appropriate methods to treat the disease. These treatment pathways are not dichotomous – strictly biomedical or Ayurvedic/traditional - but fall somewhere between the two healing styles on a spectrum.

This is a fascinating topic, brought to life by the methods employed. With emerging diseases, there is a tendency to focus only on biomedical metrics – disease presentation, progression, prevalence, etiology, etc – and very little attention is paid to the lived experience of those with the disease as the authors and their participants rightly point out. Approaching CKDu from this perspective is certainly beneficial and can add to the emerging existing knowledge around it.

I do feel, however, that the authors sold themselves short. They did not explicitly mention the (presumable) social science/qualitative gaps in CKDu literature. They do well to situate the scale of the issue (incidence rates and death) and link these with the cases in other countries, but they neglect to focus on what other social scientists have said/done, just that this area is limited. It would be useful to draw on their work and explicitly situate your own, demonstrating how it speaks to them but is distinct.

Methods

This section should be clearer. You state that you are using a transcendental phenomenological approach but you do not define what this is and how you use it. It would be useful to take a few sentences to explain to readers unfamiliar with this approach. It would also be beneficial to mention specifics regarding fieldwork. As there were two study sites, how was the time divided? Did the primary researcher split their time evenly? Were the research assistants based one in each location, or did they change halfway through? Who took fieldnotes and did you have meetings to discuss these? Or were the RAs only recruited for assisting with interviews/focus group discussions? Were they locals from the area or did they come from another part of the country (i.e. from a university?)

Specific comments

Terms

Elderly people/senior citizen – it is usually good practice to call these ‘older adults’

Patients – This feels quite a medical term to me (which is the opposite of what you are aiming for!) I would suggest ‘participants’. Also, they are not all ‘patients’ if you also spoke to healthcare providers.

Glomerular Filtration Rate-GFR (line 238) – I think some explanation of what this is would be useful, I was not expecting such a technical term to be used so some brief context would be beneficial.

Chena cultivation (line 323) – I am unfamiliar with this term and there is no explanation for what it is/entails.

You begin with saying ‘three decades’ but in the second have you state ‘two decades’ (e.g. line 505). Make sure these are consistent.

Ideas

Humanitarian concerns – I was very eager to read this section as my understanding of humanitarian actions are of someone providing aid/support etc. Looking back, I understand why you used this term, but I was expecting some organization in Sri Lanka to be offering support to those in these hyper-endemic areas. Consider rephrasing this to avoid future reader confusion such I as I had.

Snakebites (lines 372-376) – This is a small paragraph and it could certainly use some expansion. Why did people relate CKDu to snakebites? Just because more men were afflicted than women? How did they think the snakebites caused it? Something in the venom? This section could be richer in detail.

Stigma – Very interesting that stigma may be associated with the disease. Is there any other literature corroborating this? What are the negative social consequences that you mention? Do people avoid families where a member has this disease? You also start with stigma and end with karma/bad luck. How are these ideas related in the minds of your participants?

Paragraph beginning line 638 – This needs rewritten. There are several concepts in it and it is unclear how they all relate.

Line 693-695. You mention an ‘increasing demand for complementary medicine in the contemporary society’ but cite literature from 1999. Please find a more recent citation as this is dated and may have changed.

Psychosocial – This is an idea throughout but I don’t readily see how it is relevant to your discussion. You advocate for psychosocial counseling services to be included in a clinical setting but I am not sure what purpose you see this as serving. I would assume psychosocial counseling would be for those suffering from some type of mental distress/illness like depression or anxiety. This is not mentioned in any of your subthemes or in any of the participants’ narratives. Be explicit as to why this is important/relevant to what you have written, or rethink including this concept/wording it in this way.

Conclusion

Overall, it has a promising premise with the value and insight that ethnography offers from the perspective of 'lay' community members. It also highlights immediate gaps that could readily be addressed - providing proper information on the disease/progression, opening up dialogues about using different treatment models in conjunction, etc. However, some of the terms used need more thought and reflection. Furthermore, the English in the article needs reviewed as it impacts on the readability of the article.

Author Response

Attached the revised manuscript with track changes for reference.  Please follow the given table with Amendment number/s in the document to find the response to relevant comment/s   

Reviewer 2 Report

This paper examines the lay perception of CKDu in two study areas in Sri Lanka. This is a hither to neglected topic in the expanding body of literature primarily based on biomedical approaches. Therefore this paper calls for serious consideration. However, the article is poorly theorized, methodologically problematic and replete with language problems. The author(s) state that the article uses transcendental phenomenological approach, but the body of the article does not give any examples of the application of the approach. The methodology is described as ethnographic, but the research procedures followed do not conform to the conventional ethnographic procedures such as detailed study of one or more selected communities. Instead the study focus on two sites that are large territories with diverse population groups in them.  The popular interpretation of the disease in the two locations appear to be influenced by a variety of knowledge systems, biomedicine, ayurveda and indigenous medicine and media portrayals of the disease, but the reader do not get a clear understanding of how the lay people deal with possible contradictions among the different epistemologies. The paper requires substantial revision focusing on all these issues raised.     

Author Response

Please find the attached revised manuscript with responses. Please refer to the table of reviewers' comments and Amendment number to find the relevant comment 

Reviewer 3 Report

This is a very interesting paper and is line with the special issue call. I have the follow concerns:

  1. This paper does need a really thorough English language edit.
  2. methods: how was the participant observation data collected/recorded? How was purposive sampling applied? Who is in the FGDs (this is not clear in the methods, only the results); how was trustworthiness/quality ensure throughout the research process (who is the I in the interviewer?)
  3. Ethics: My ethical concern is around confidentiality with reporting of exact ages of participants - I wondered if this could make some participants identifiable to doctors/teams who treated them. Putting in age brackets rather than exact ages would help here.
  4. Findings - Be consistent whether you italicise the  quotes or not; “uncomfortable and unfulfilling life” – participants words or your analysis.
  5. The paper calls for more partnerships and integration across biomedicine/traditional and community perspectives and a more holistic conceptual framework with the community at the centre. There is a growing body of work on this and it would be useful to state the findings here provide practical recommendations on how these partnerships could be fostered in the Sri Lankan context.

Author Response

Herewith I attached the revised manuscript for your perusal. Please refer to the table of reviewer's comments and to find the relevant amendment 

Round 2

Reviewer 1 Report

The authors have clearly put time and effort into addressing the previous comments. This is a much-improved and stronger version. I am satisfied with the changes made. This is a very interesting and timely study that will inform future qualitative research on CKDu within Sri Lanka.

Reviewer 2 Report

The revised version of the essay addresses the issues raised in the previous review. However, application of the phenomenological approach in the analysis remains rather limited.